# Modeling the Kinetic Behavior of Reactive Oxygen Species with Cerium Dioxide Nanoparticles

**DOI:** 10.3390/biom9090447

**Published:** 2019-09-04

**Authors:** Kenneth Reed, Nathan Bush, Zachary Burns, Gwendolyn Doherty, Thomas Foley, Matthew Milone, Kara L. Maki, Michael Cromer

**Affiliations:** 1School of Chemistry and Materials Science, Rochester institute of Technology, 1 Lomb Memorial Dr, Rochester, NY 14623, USA; 2School of Mathematical Sciences, Rochester institute of Technology, 1 Lomb Memorial Dr, Rochester, NY 14623, USA

**Keywords:** cerium dioxide, reactive oxygen species, ROS, nanotechnology, kinetics, catalase, dismutation, steady-state kinetics, time-dependent kinetics, system of differential equations

## Abstract

The world of medicinal therapies has been historically, and remains to be, dominated by the use of elegant organic molecular structures. Now, a novel medical treatment is emerging based on CeO_2_ nano-crystals that are discrete clusters of a few hundred atoms. This development is generating a great deal of exciting and promising research activity, as evidenced by this Special Issue of *Biomolecules*. In this paper, we provide both a steady-state and time-dependent mathematical description of a sequence of reactions: superoxide generation, superoxide dismutase, and hydrogen peroxide catalase and ceria regeneration. This sequence describes the reactive oxygen species (ROS); superoxide, O_2_^–^, molecular oxygen, O_2_, hydroxide ion OH^–^ and hydrogen peroxide, H_2_O_2_, interacting with the Ce^3+,^ and Ce^4+^ surface cations of nanoparticle ceria, CeO_2_. Particular emphasis is placed on the predicted time-dependent role of the Ce^3+^/Ce^4+^ ratio within the crystal. The net reaction is succinctly described as: H_2_O_2_ + 2O_2_^–^ + 2H^+^ → 2H_2_O + 2O_2_. The chemical equations and mathematical treatment appears to align well with several critical in vivo observations such as; direct and specific superoxide dismutase (SOD), ROS control, catalytic regeneration, ceria self-regulation and self-limiting behavior. However, in contrast to experimental observations, the model predicts that the 4+ ceric ion state is the key SOD agent. Future work is suggested based on these calculations.

## 1. Introduction

The rare earth metal oxide, CeO_2_, commonly called Ceria, is one of those versatile and powerful redox catalysts whose widespread and expanding utility to modern society has been the subject of many studies. The applications space is diverse; optical lens polishing, chemical/mechanical planarization of semiconductor wafers, diesel oxidation catalyst, catalytic converters for internal combustion engines, solid oxide fuel cells and sensors [1,2]. It is a fortunate circumstance that the presence of ceria in the earth’s crust is actually not that rare.

It might seem odd at first glance that this inorganic catalyst might play such a significant role in human biology [3,4]. This role may in fact be transformational in the control of intractable human central nervous diseases like, Multiple Sclerosis [5], ALS [6], Eschemic Stroke [7,8], Parkinson’s Disease [9], perhaps even Alzheimer’s Disease [10]. The unifying chemistry that these pathologies share, to varying degrees, is the driving force of cellular damage by excess reactive oxygen species (ROS) [11,12]. For a particular ROS superoxide O_2_^–^, ceria has shown the potential to act as a superoxide dismutase mimetic (an enzyme-like material converting superoxide to hydrogen peroxide and O_2_) and a catalase mimetic (an enzyme-like material that disproportionates hydrogen peroxide) [13,14,15,16,17,18,19]. The word “nanozyme” (a portmanteau of nanomaterial and artificial enzyme) is another way of labeling ceria based on its biological action [20].

Pivotal to understanding the behavior of this solid-state/liquid interface chemistry is the realization that oxygen vacancy formation [21] (i.e., the removal of an oxygen atom from the crystal lattice that results in two Ce^4+^ ions becoming reduced to Ce^3+^) and the time evolution of the concentration ratio of these ions, Ce^3+/^Ce^4+^, is the thermodynamic force for the extent and rate of control of ROS concentration in the presence of this unique metal oxide.

The literature describing the mechanisms and kinetics of ROS interacting with ceria is somewhat sparse but what exists, is quite intriguing.

Calander et al. modeled the pharmokinetic distribution of both 5 and 30 nm citrate-coated ceria in various organs (spleen, liver, brain, in addition to blood) and concluded that the biokinetics depended largely upon the dosage and mode of delivery [22]. The time axes spanned days so that very little information was discernable on the order of hours.

Baldim et al. examine the Super Oxide Dismutase, SOD, and Catalase, CAT-like catalytic behavior of a ceria size series, 4.5, 7.5, 23, and 28 nm and find that the smaller particles show enhanced SOD effects due to the increase in Ce^3+^ surface area concentration, while for the catalase activity, adsorption of H_2_O_2_ at specific surface sites must be taken into account [23].

Mitchell, Abboud and Christou explore the opposite end of the ceria size spectrum with an upper limit of ~ 1.8 nm and detail the H^+^ and other various ligand binding sites as a function of ceria surface Miller Index, as well as the location of Ce^3+^ and oxygen vacancies [24].

Celardo, Pedersen, Traversa and Ghibelli, provide a very plausible model of a reaction mechanism schematic on the surface of a cerium dioxide nanoparticle for the dismutation of superoxide and another model of the reaction mechanism for the complete dismutation of hydrogen peroxide on a similar crystal surface [25]. No kinetic data are provided nor is there any test of both proposed mechanisms.

In the following work, we provide a self-consistent system of chemical reactions with their associated mathematical equations that describe how the dominant reactive oxygen species, superoxide, is kinetically controlled when it interacts with both oxidized and reduced ceria cations in CeO_2_ nano-crystals. This very simple, foundational model, neglects the effects of other reactive oxygen species such as hydroxyl radical, nitrous oxide and peroxynitrite, while focusing on an acknowledged key player, superoxide (implicated in, for example, ALS and Cystic Fibrosis). Diffusion effects of the various reacting species are also neglected at this first level of treatment. The goal is to construct a mathematical description of an elementary set of chemical reactions that captures the critical features of some in vivo and in vitro observations and to see where the model is at variance with actual experiments.

## 2. Mathematical Models

The reactions we are concerned with are described as follows:(1)$$H_{2}O_{2}+O_{2}+2OH^{-}\to 2O_{2}^{-}+2H_{2}O\;k_{1}$$

(2)$$Ce^{4+}+O_{2}^{-}\to Ce^{3+}+O_{2}\;k_{2}$$

(3)$$Ce^{3+}+O_{2}^{-}+2H^{+}\to Ce^{4+}+H_{2}O_{2}\;k_{3}$$

(4)$$H_{2}O_{2}+2Ce^{3+}+2H^{+}\to 2Ce^{4+}+2H_{2}O\;k_{4}\;$$

Wherein, reaction (1) represents the generation of [O_2_^–^], causing the subsequent disease state in the body. Reactions (2) and (3) describe the superoxide dismutation reaction that destroys $O_{2}^{-}$ and recycles Ce^3+^ to Ce^4+^. Reaction (4) describes the hydrogen peroxide catalase reaction and regeneration of Ce^4+^. Note that these reactions are, individually and overall, balanced as they obey both the conservation of mass and conservation of charge. They represent one posssible complete cycle of a disease progression from generation to destruction of the ROS species, O_2_^–^ and $H_{2}O_{2}$.

Also of interest is the dual role of hydrogen peroxide. In basic media, it is a reducing agent and in acidic media, it is an oxidant. Thus, the role of water (a constant in concentration due to its great excess) is manifested through its dissociation constant providing locally, both hydroxyl and hydronium ions at various surface sites of the crystal.

To describe the species at the crystal surface, we simply denote them as Ce^4+^ and Ce^3+^, while the other species are in some continuous phase, freely diffusing.

One possible representation of a small ceria nano-crystal Ce_79_O_160_ is given below (Figure 1), wherein the red atoms represent oxygen and the yellow atoms represent cerium.

The figure on the left represents an oxygen atom surface vacancy (two color beach ball in a {111} lattice plane that is shown by a cyan arrow), while the right figure shows that the oxygen vacancy has migrated to the lowest energy lattice position that happens to be in the crystal interior. The Ce^3+^ ions relocate to the crystal surface where they can better be spatially accommodated now that their ionic radius has increased [1].

The reduction of two +4 cerium ions in the lattice occurs when an oxygen lattice ion becomes fugitive from the crystal lattice, leaving an oxygen vacancy behind. This oxygen vacancy is charge-balanced by two neighboring +3 ions. It is at this oxygen vacancy site that the catalysis is thought to occur. This oxygen vacancy may very well be the reason for the specificity of the reaction sequence to superoxide, O_2_^–^ and not other ROS species such as NO, OH^.^, RO^.^.

The sum of reactions (1) through (4) is:
H_2_O_2_ + 2O_2_^–^ + 2H^+^ → 2H_2_O + 2O_2_(5)

Examination of Equation (5) reveals that indeed, the nano-ceria is acting as a catalyst, as it has, on balance, neither been produced as a by-product nor is it consumed as a reactant. Additionally, superoxide, the source of the pathenogenisis (for example, in ALS) is consumed while no toxic side products are made (therefore, it is not expected to be immunomodulatory) and the pH of the system is restored (raised from a low value in the diseased state) as hydronium ion is consumed.

We examine the energy balance of the system of these equations, using free energy, ΔG. We find:

ΔG^0^
_reaction_ =2(ΔG^0^_O2_) + 2(ΔG^0^
_H2O_) – 2(ΔG^0^
_H+_) − 2 (ΔG^0^_O2-_) − ΔG^0^_H2O2_

= 2(231.7) + 2(−237.1) − 2(0) − 2(33.8) – (−120.4)

= 42 kJ/mole = 10.04 kcal /mole

This net reaction faces a slight thermodynamic barrier but not unreasonably so.

## 3. Results

### 3.1. Steady-State Kinetics

Writing the critical differential equations for the concentration of species Ce^3+^ and Ce^4+^, we find that:$$\frac{d\left[{\mathrm{Ce}}^{4+}\right]}{\mathrm{dt}}=-k_{2}\left[{\mathrm{Ce}}^{4+}\right]\left[O_{2}^{-}\right]+k_{3}\left[{\mathrm{Ce}}^{3+}\right]\left[O_{2}^{-}\right]\left[H^{+}\right]+k_{4}\left[H_{2}O_{2}\right]\left[{\mathrm{Ce}}^{3+}\right]\left[H^{+}\right]=-\frac{d\left[{\mathrm{Ce}}^{3+}\right]}{\mathrm{dt}},$$
such that
$$\frac{d}{\mathrm{dt}}\left[{\mathrm{Ce}}^{4+}+{\mathrm{Ce}}^{3+}\right]=0$$
and
$$\left[{\mathrm{Ce}}^{4+}+{\mathrm{Ce}}^{3+}\right]=\left[{\mathrm{Ce}}^{4+}\right]+\left[{\mathrm{Ce}}^{3+}\right]=C_{t}.$$

At a steady-state solution, with d [Ce^4+^]/dt = 0,

(6)$$\frac{\left[{\mathrm{Ce}}^{3+}\right]}{\left[{\mathrm{Ce}}^{4+}\right]}=\frac{k_{2}\left[O_{2}^{-}\right]}{k_{3}\left[H^{+}\right]\left[O_{2}^{-}\right]+k_{4}\left[H^{+}\right]\left[H_{2}O_{2}\right]}.$$

Let us consider two limiting cases.

Case 1. [O_2_^–^] >> [H_2_O_2_] (the disease state): Here, [Ce^3+^]/[Ce^4+^] = k_2_/k_3_[H^+^] and the ceria therapy is dominated by the ratio of dismutation rates (i.e., oxygen vacancy formation and destruction). Note, also by rearrangement, we see from this that [Ce^3+^][H^+^]k_3_ = [Ce^4+^]k_2_, the ceria redox system is in balance. That is to say, cerous and ceric ions are being created and destroyed in balance (conserved) and the system is self regulating.

Case 2. [H_2_O_2_] >> [O_2_^–^] (the healthy state): Now [Ce^3+^]/[Ce^4+^] = k_2_[O_2_^–^]/k_4_[H^+^][H_2_O_2_ ] and the 4+ state dominates and not much occurs (i.e., no dismutation process and the ceria system has shut itself down); it is self-limiting, which is beneficial since it is not needed.

Both of these conditions—self-regulating and self-limiting—are remarkable features not found in everyday chemical therapies.

### 3.2. Time-Dependent Kinetics

For ease of notation and mathematical tractability, we remap the identity of each of the chemical species into the single symbol, most easily identified with that chemical species. For example, the concentration of hydrogen peroxide, $\left[H_{2}O_{2}\right]$, becomes the letter P; oxygen, $\left[O_{2}\right]$, becomes O; $\left[{\mathrm{Ce}}^{3+}\right]$ is denoted C; while $\left[{\mathrm{Ce}}^{4+}\right]$ is F, and H is the hydronium ion H^+^, and D is hydroxide ion OH^–^. Concentrations have their normal molarity description.

The Law of Mass Action is used to formulate a set of differential equations that describes the concentrations of all chemicals of interest as a function of time. The total rate of reaction for superoxide, S, is the time rate of change of S and is equal to the rate of reactions producing S minus those reactions depleting S or

$$\frac{\mathrm{dS}}{\mathrm{dt}}=2k_{1}{\mathrm{POD}}^{2}-k_{2}\mathrm{FS}-k_{3}{\mathrm{CSH}}^{2}$$

The equations are dimensionally homogeneous.

Because the Ce^3+^ and Ce^4+^ cerium ions are locked into a crystal lattice, their sum total must be a constant. This requires that
C*_t_* = C + F,
where C*_t_* is a constant total concentration of ceria.

The initial concentration of +3 and +4 ceria and their initial ratio will depend on a number of factors that can be experimentally manipulated, such as crystal size, surface functionalization, and the number and charge of any dopant ions in the ceria lattice, such as La^3+^. Experimental determination of that ratio is fraught with subtlety and nuance [1].

Further simplifications in the system of equations can be made once we note that none of the equations depend on *W*, i.e., [H_2_O], since it only appears as a product of the reactions and is present in great excess. Thus, its concentration does not change appreciably with time. The presence of water is manifested through its dissociation constant, which provides both H^+^ and OH^–^ concentrations, which are critical to the overall set of reactions. Taking these facts into account, we simplify the system into the following set of equations:$$\frac{\mathrm{dP}}{\mathrm{dt}}=-k_{1}{\mathrm{POD}}^{2}+k_{3}{\mathrm{CSH}}^{2}-k_{4}{\mathrm{PC}}^{2}H^{2}$$

$$\frac{\mathrm{dO}}{\mathrm{dt}}=-k_{1}{\mathrm{POD}}^{2}+k_{2}\left(C_{t}-C\right)S$$

$$\frac{\mathrm{dD}}{\mathrm{dt}}=-2k_{1}{\mathrm{POD}}^{2}$$

$$\frac{\mathrm{dS}}{\mathrm{dt}}=2k_{1}{\mathrm{POD}}^{2}-k_{2}\left(C_{t}-C\right)S-k_{3}{\mathrm{CSH}}^{2}$$

$$\frac{\mathrm{dC}}{\mathrm{dt}}=k_{2}\left(C_{t}-C\right)S-k_{3}{\mathrm{CSH}}^{2}-2k_{4}{\mathrm{PC}}^{2}H^{2}$$

$$\frac{\mathrm{dH}}{\mathrm{dt}}=-2{\mathrm{CH}}^{2}\left(k_{3}S+k_{4}\mathrm{PC}\right)$$

We also assumed that no reactants are being introduced or removed after *t* = 0. For example, we assumed that ${\mathrm{OH}}^{-}\;$is not being continuously produced and that no ceria is being filtered out. This simplified the model, but affects the frequency and dosages of ceria needed to keep the concentrations of potentially harmful species below dangerous levels.

### 3.3. Non-Dimensionalization

After finding a simplified system, we non-dimensionalize where each variable’s characteristic scale is denoted with a subscript of “*c*”. For example, the scale for oxygen is $O_{c}$ and let $\hat{O}=O/O_{c}$. In this manner, all instances of O can be replaced with $O_{c}\hat{O}$.

In order to reduce the total number of parameters, scales were chosen that use ratios of rate constants and the total ceria concentration to reduce the number of coefficients that simplify the problem. The scales we found are:$$t_{c}={\left(\frac{k_{4}}{k_{1}}\right)}^{2}\frac{1}{C_{t}k_{3}}$$

$$F_{c}=C_{c}=S_{c}=D_{c}=C_{t}$$

$$P_{c}=\frac{k_{3}}{k_{4}}$$

$$O_{c}=H_{c}=\frac{k_{1}}{k_{4}}$$

The resulting system is:$$\frac{d\;\hat{P}}{d\;\hat{t}}=\frac{k_{4}}{k_{3}}C_{t}\left({\hat{C}\hat{S}\hat{H}}^{2}-{\hat{P}\hat{O}\hat{D}}^{2}-{\hat{P}\hat{C}}^{2}{\hat{H}}^{2}\right)$$

$$\frac{d\hat{O}}{d\hat{t}}=\frac{k_{4}}{k_{1}}C_{t}\left(\frac{k_{2}}{k_{3}}{\left(\frac{k_{4}}{k_{1}}\right)}^{2}\left(1-\hat{C}\right)\hat{S}-\hat{P}\hat{O}\hat{D^{2}}\right)$$

$$\frac{d\hat{D}}{d\hat{t}}=-2{\hat{P}\hat{O}\hat{D}}^{2}$$

$$\frac{d\hat{S}}{d\hat{t}}=2{\hat{P}\hat{O}\hat{D}}^{2}-\frac{k_{2}}{k_{3}}{\left(\frac{k_{4}}{k_{1}}\right)}^{2}\left(1-\hat{C}\right)\hat{S}-{\hat{C}\hat{S}\hat{H}}^{2}$$

$$\frac{d\hat{C}}{d\hat{t}}=\frac{k_{2}}{k_{3}}{\left(\frac{k_{4}}{k_{1}}\right)}^{2}\left(1-\hat{C}\right)\hat{S}-{\hat{C}\hat{S}\hat{H}}^{2}-2{\hat{P}\hat{C}}^{2}{\hat{H}}^{2}$$

$$\frac{d\hat{H}}{d\hat{t}}=2\left(\frac{k_{4}}{k_{1}}\right)C_{t}-\left({\hat{C}\hat{S}\hat{H}}^{2}+{\hat{P}\hat{C}}^{2}{\hat{H}}^{2}\right)$$

From here on, all references to the equations and variables refer to their non-dimensional form. When referring to a variable, we interpret it as the variable relative to its scale. For example, superoxide represents the concentration of superoxide relative to the total concentration of ceria, peroxide represents the concentration of peroxide relative to the ratio of $k_{3}$ to $k_{4}$, and so on.

## 4. Results

### Time-Dependent Kinetics

We are interested in both the transient and steady-state behavior of the mathematical model. For the latter, we set d/dt = 0 in the above equations and find admissible solutions for the nonlinear system of equations. In this manner, we seek analytical, long-time solutions of the dynamical system, which provides the possible states to which the transient equations will evolve.

Let (P*,O*,D*,S*,C*,H*) represent the steady-state solutions to the model. We have found eight possible solutions, three of which correspond to S* not necessarily 0, and five for which S* is identical to 0, which is the ideal scenario since S* corresponds to excess O_2_^–^. The three solutions, with S* nonzero, are: (P*,O*,0,S*,1,0), (0,O*,D*,S*,1,0), and (P*,0,D*,S*,1,0). The remaining five “ideal” solutions are: (P*,0,D*,0,0,H*), (P*,O*,0,0,0,H*), (P*,0,D*,0,C*,0), (P*,O*,0,0,C*,0), and (0,O*,D*,0,C*,H*).

There are several important things to note. First, C* = 1 in all of the non-ideal cases, which means that the total ceria concentration at steady-state is comprised of Ce^3+^. In the ideal cases, we note that the concentrations of Ce^3+^ and Ce^4+^ can either be 1, 0, or anywhere in between, given by the free parameter C*. Thus, it is possible that a steady solution with O_2_^–^ = 0 could correspond to Ce^3+^ = C_t_, i.e., all ceria is Ce ^3+^, or all Ce ^4+^.

In the chemical reactions, Ce^3+^ and Ce^4+^ convert between each other. Thus, in order to have a better chance at avoiding the non-ideal cases, and to allow as much flexibility as possible in the selection of steady states where O_2_^–^ goes to 0, a possible ideal initial dosage would be one completely comprised of Ce^4+^, i.e., starting as far away as possible from the non-ideal cases.

We approximated the system of differential equations with MATLAB’s ode45 integrator that implements the Runge–Kutta method.

The main goal is to eliminate as much of the excess superoxide in the system as possible. In all results below, we have used $k_{4}/k_{1}=k_{2}/k_{3}=k_{4}/k_{3}=C_{t}=1$. In the first case, we considered what would happen if the only ceria was present purely in the form of cerous ion, Ce^3+^; therefore, the crystal would be completely Ce_2_O_3_ or in the case of ceric ion Ce^4+^, CeO_2_. What would the results look like in these scenarios?

Figure 2 shows that the species Ce^3+^ does not reduce the amount of O_2_^–^ at all. There is a slight increase in the amount of superoxide before it decreases back to its initial amount as the amount of Ce^3+^ recovers (i.e., increases) back to its original level. The Ce^3+^ is somewhat ineffective in controlling O_2_^–^.

Now, consider Figure 3, where only Ce^4+^ is present in the crystal, i.e., CeO_2_. We see that all of the superoxide is eventually eliminated. Although there is a similar small increase in the amount of the superoxide, its concentration diminishes along with $H^{+}.\;$As the system reaches a steady state, all of the ceria in the system is converted to Ce^3+^, while none of the Ce^4+^ remains. The system is very active in controlling superoxide, unlike the results in Figure 2.

In both Figure 2 and Figure 3, there is a slight reduction in the amount of H_2_O_2_ (not shown), as well as an increase in O_2_. From these initial plots, it appears that the ceric ion is much more effective than the cerous ion at reducing the superoxide anion concentration.

From this knowledge of the importance of the +4 state, in the next few cases, we consider starting with only an initial concentration of Ce^4+^. As was calculated in the non-dimensional equations, many of the terms include ratios of rate constants. We now consider what would happen if one of these two rate constants were greater than the other and what would happen if those reactions occurred at different rates.

In Figure 4, we see what happens when *k*_1_ is the dominating rate constant, that is, the production of superoxide overpowers the rest of the system and a serious disease state exists.

The plots in Figure 4 and Figure 5 display diametrically opposite results. In Figure 4, (large excess superoxide generation rate) the redox couple Ce^3+/^Ce^4+^ does not seem to be responding over time and while $H^{+}$ is decreasing, the hydroxide concentration must be increasing. This corresponds to an excess of superoxide and the severe disease state.

Figure 5 shows a much faster and better rate of control of superoxide and a very active Ce^3+/^Ce^4+^ redox couple.

The next few figures look into the dosage amount of ceria that should be used in relation to the amount of excess superoxide that is present. Does introducing a much greater amount of ceria have an overall effect on the rate of control of superoxide? What about the opposite scenario, in which a smaller amount of ceria is initially used? Will the excess superoxide be controlled in a reasonable time scale? Figure 6 deals with starting with a larger amount of Ce^+4^ when superoxide is initially present.

In Figure 6, it is interesting to note that both the superoxide and hydrogen peroxide are effectively reduced in almost the same time as in Figure 3, while leaving a balance of the +3 and +4 cerium, which one would expect due to the initial excess. Experimentally, it is observed that an increase in ceria dosage results in a greater reduction of the severity of the disease [26].

Figure 7 and Figure 8 also deal with the case of under dosing the amount of ceria but with particular regard to the amount of $H^{+}\;$also present in the system and show some particularly intriguing results. With only Ce^3+^ initially present (Figure 7), superoxide is not eliminated. With only Ce^4+^ initially present (Figure 8), superoxide is eliminated.

It seems that if the amount of H^+^ present in the system is approximately equal to the amount of O_2_^–^, a large excess of O_2_^–^ will still be eliminated. Both of these species decrease at the same rate, which may be attributed to how H^+^ is used to help eliminate O_2_^–^, as well as dismutate the ceria ions. Without a sufficient amount of $H^{+}$, the Ce^3+^ ions cannot be converted into Ce^4+^.

Once again, we see the need for CeO_2_, and not Ce_2_O_3_, initially as excess superoxide increases and ultimately, persists. There is also an increase in O_2_ and H_2_O_2_ accompanying the destruction of superoxide. Another result that is similar to Figure 3 is that all the ceria converts to Ce^3+^ at the steady state. If this pattern is consistent, and we are able to closely monitor and control the amount $H^{+}$, then large amounts of O_2_^–^ could be reduced with only a amount of ceria needed.

The dynamic ratio of the +3 to +4 ion is very dependent on the initial starting conditions: exclusive +3 initially, will eventually return to this state after undergoing oxidation, while an exclusive +4 state will all be reduced to Ce^+3^ (Figure 9).

In this last plot, Figure 10, we see another result that came about through testing by varying initial conditions. Since the reaction that dictates the generation of O_2_^–^
is driven by the presence of ${\mathrm{OH}}^{-}$, we wanted to see if an increased presence of that species would alter the effectiveness of ceria. Figure 10 leads us to believe that a large amount of ${\mathrm{OH}}^{-}\;$does in fact change the rate at which excess O_2_^–^ is reduced. The amount nearly doubles when it reaches a steady state. This indicates that the species ${\mathrm{OH}}^{-}\;$is also vital to keep track of in the system. Instead of assisting in the process of removing excess O_2_^–^ like the $H^{+}$ did, it appears to have an adverse effect on the problem. The ceria cannot counteract the increased generation of O_2_^–^, leading to an ineffective resolution of the superoxide pathology.

## 5. Discussion

The main goal for the model was to show the qualitative role of ${\mathrm{Ce}}^{4+}$ and ${\mathrm{Ce}}^{3+}$ in reducing the amount of $O_{2}^{-}$ in the body. We explored various initial concentrations of ${\mathrm{Ce}}^{4+}$ and ${\mathrm{Ce}}^{3+}$. From our simulations, we found that when there was much more ${\mathrm{Ce}}^{4+}$ than ${\mathrm{Ce}}^{3+}$, superoxide would decrease to a steady state of 0 (i.e., what we define as no excess superoxide, not necessarily zero superoxide). As ${\mathrm{Ce}}^{4+}$ was converted into ${\mathrm{Ce}}^{3+}$, the superoxide levels dropped. We infer that ${\mathrm{Ce}}^{4+}$ should be higher than ${\mathrm{Ce}}^{3+}$, as ${\mathrm{Ce}}^{4+}$ seems to be the critical chemical species that reduces the superoxide. Our results are not in agreement with the data of Pirmohamed [17] on 5–10 nanometer ceria particles nor that of Heckert et al. [18], that separately categorize 3+ ceria as a SOD-like mimetic and 4+ ceria as a catalase-like mimetic. This discrepancy is primarily due to the dual role we have assigned the 3+ state in both destroying O_2_^–^ (SOD or dismutase-like which is Equation (3)) and in reducing hydrogen peroxide (catalyse-like mimetic, Equation (4)). The prevailing general belief that the 3+ state is critical to addressing super oxide dismutation chemistry, while the 4+ state is hydrogen-peroxide-catalase-dominant may stem from the nature of the ceria crystals being studied. Relatively large (10–15 nm) highly agglomerated ceria crystals will in fact be highly depleted in the 3+ ion concentration as has been shown by Hailstone [26] and, therefore, under these circumstances, the ${\mathrm{Ce}}^{3+}$ species may be rate limiting.

We also determined that the value of $k_{1}$ had an effect on how quickly the chemicals reached their steady states. The reaction rate governed by, $k_{1}$, can be sped up or slowed down by using excess peroxide and base. When $k_{1}$ was set to 5-times higher than $k_{2}$, $k_{3}$, and $k_{4}$, the equations reached their steady states much more slowly. In a sense, this very severe disease state is overwhelming the recovery reactions governed by $k_{2}$, $k_{3}$, and $k_{4}$ and protracting their time scales. Eventually, all O_2_^–^ is converted to O_2_. When $k_{1}$ is lower than $k_{2}$, $k_{3},$ and $k_{4}$, the reactions reach a steady state much quicker and O_2_^–^ never goes to zero in this scenario. These two extreme cases of the relative rate constants of generation and recovery from the disease state reflect on what could be interpreted as a dose–response behavior that has been observed previously [27,28].

It was also found that the amount of $H^{+}$ helped to play a factor in reducing O_2_^–^ excess. This is immediately evident by inspection of the overall equation: H_2_O_2_ + 2O_2_^–^ + 2H^+^ → 2H_2_O + 2O_2_.

Introducing ceria with the correct amount of $H^{+}$ present can help in eliminating large amounts of O_2_^–^. This trend could possibly be very cost effective since much smaller amounts of ${\mathrm{Ce}}^{4+}$ can eliminate O_2_^–^ with minimal negative effects although one would not want to get outside the range of normal physiological pH values (6.7–7.2).

The model was constructed to the qualitative behavior of the chemical reactions. We note that the model does not account for biological processes that modify the chemical composition of the environment in which the model is set. For example, the circulatory system is constantly introducing oxygen to the reaction environment, and mitochondria are constantly converting oxygen into metabolic products. Both of these phenomena become significant above the time frame of seconds, but their precise effects on the chemical model are unknown. These effects, as well as species diffusion, incorporation of other reactive oxygen species, as well as reactive nitrogen species, could be the subject of future model refinements.

According to the model, the production of superoxide is limited by the initial concentration of hydroxide, and the destruction of superoxide is limited by the initial concentration of hydrogen ions (as these generate the ${\mathrm{Ce}}^{4+}$ state). However, we know that the concentrations of these two chemicals can never drop to arbitrarily low levels, because $[H^{+}]\left[{\mathrm{OH}}^{-}\right]=10^{-14}$. There may be a local, temporary depletion of these ions at certain surface sites but in a predominantly aqueous environment, the bulk equilibrium concentration of hydroxyl and hydronium ions will be maintained.

## 6. Conclusions

This self-consistent system of equations and the resultant net equation seem to connect quite strongly, qualitatively with observations made in the laboratory. This is both gratifying and encouraging. To wit:When some ${\mathrm{CeO}}_{2}$ is oxidized or reduced, this treatment has little effect on sparing. That is because both ${\mathrm{Ce}}^{3+}$ and ${\mathrm{Ce}}^{4+}$ consume $O_{2}^{-}.$Hydrogen peroxide is consumed, consistent with observations that ceria is working on an ALS murine model.If the antagonist (hydrogen peroxide) goes away (i.e., the organism is healthy), the entire reaction sequence shuts down (i.e., ceria is not immunosuppressant, nor does it interfere with mitochondrial function).The system is catalytic (regenerative and not sacrificial).The system is both self-limiting and self-balancing.It appears to directly control reactive oxygen species and specifically superoxide, perhaps due to the nature of the oxygen vacancies within the crystal.Within the framework of the reaction sequence as described, the dominant ions are the cerium 4+ state and H+ initially but in totality, the chemical reaction system requires both 4+ and 3+ cerium ions, which are constantly in balance and equilibrating.

## Figures and Tables

**Figure 1 biomolecules-09-00447-f001:**
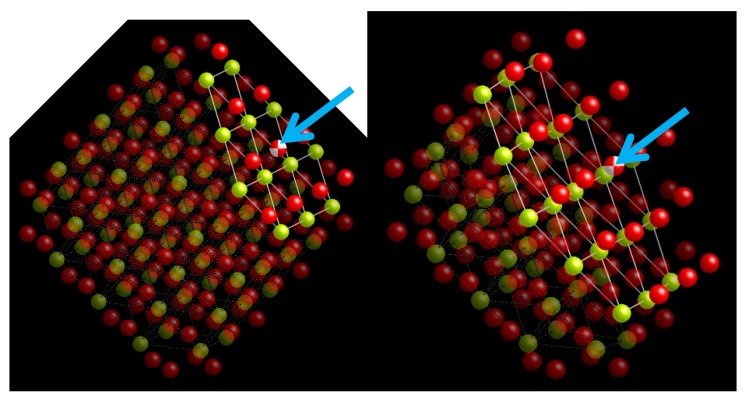
Creation of a surface oxygen vacancy (shown by a cyan arrows) that subsequently migrates to the crystal interior where it is energetically more stable (Personal communication, Professor Alastair Cormack, Alfred University, Alfred, New York).

**Figure 2 biomolecules-09-00447-f002:**
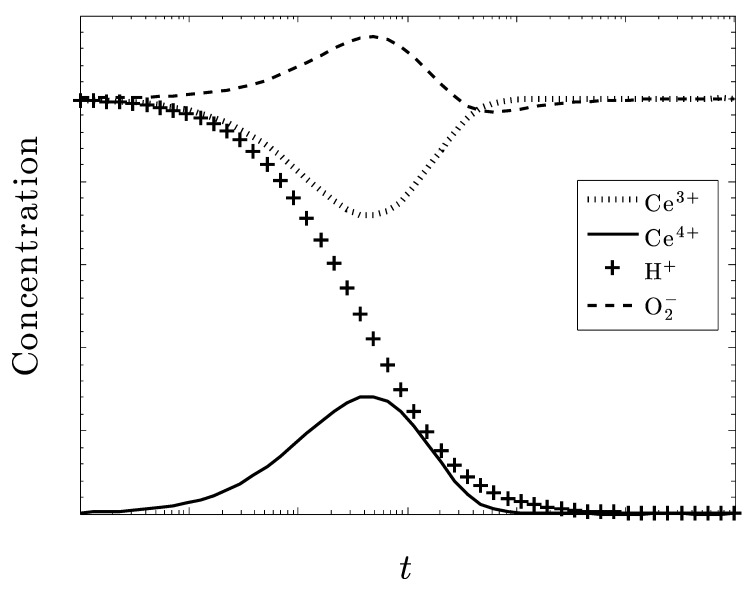
Only Ce^3+^ ion is present in the Ce_2_O_3_ crystal. All other species are equivalent as are their respective rate constants. There is a slight increase in the superoxide concentration that then returns to its original level.

**Figure 3 biomolecules-09-00447-f003:**
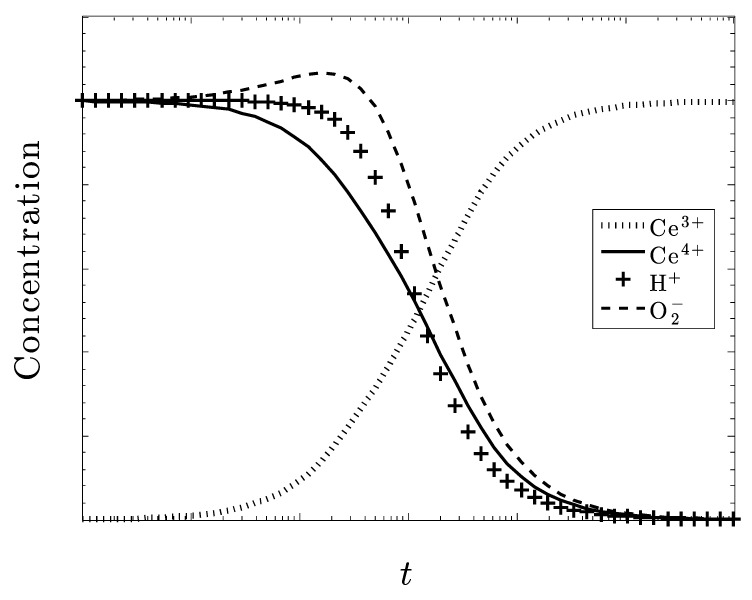
Only Ce^4+^ ion present in the CeO_2_ crystal. All other species are equivalent as are their respective rate constants. The superoxide concentration is completely eliminated as the system converts to Ce_2_O_3_.

**Figure 4 biomolecules-09-00447-f004:**
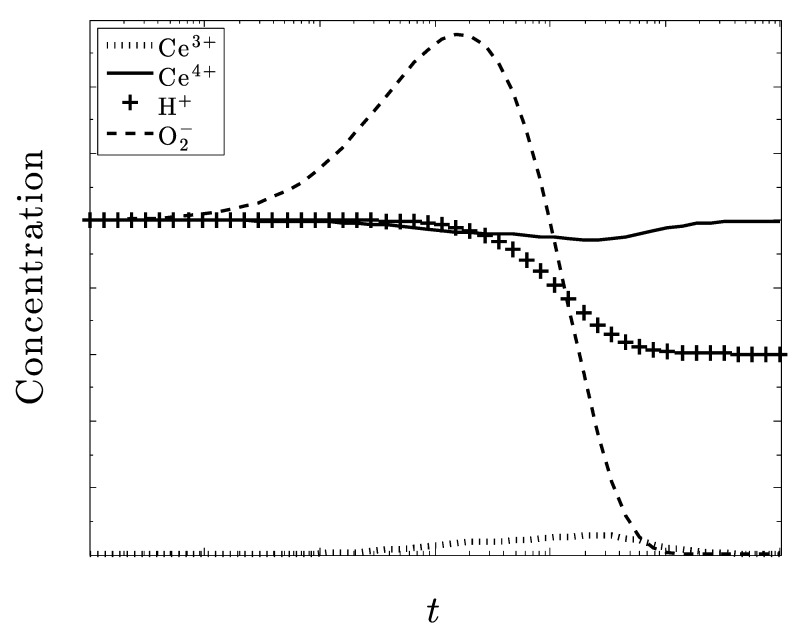
Now k_1_ is fixed at five-times greater than k_2_, k_3_ and k_4_. The other species concentrations are equal, and the system again starts out as CeO_2_.

**Figure 5 biomolecules-09-00447-f005:**
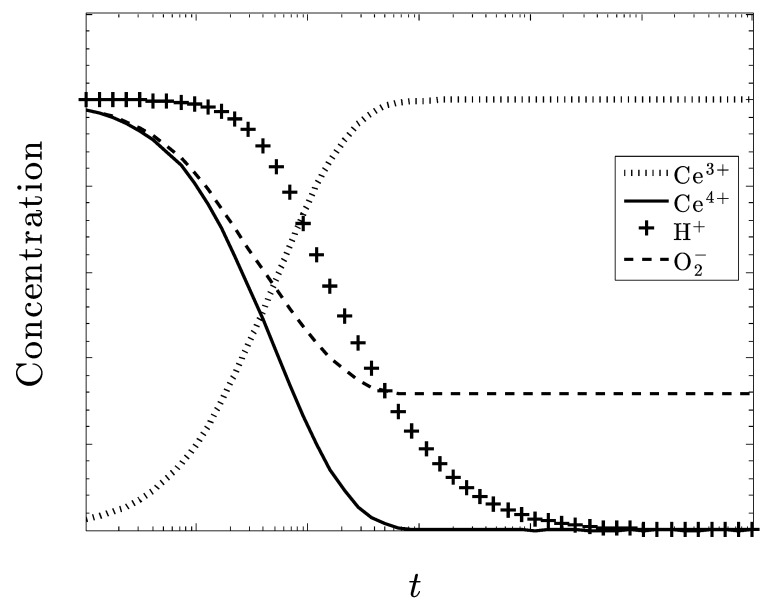
k_4_ dominates and the recovery from low pH and excess hydrogen peroxide presumably eliminates the pathenogenisis. Note, that the superoxide is not completely eliminated.

**Figure 6 biomolecules-09-00447-f006:**
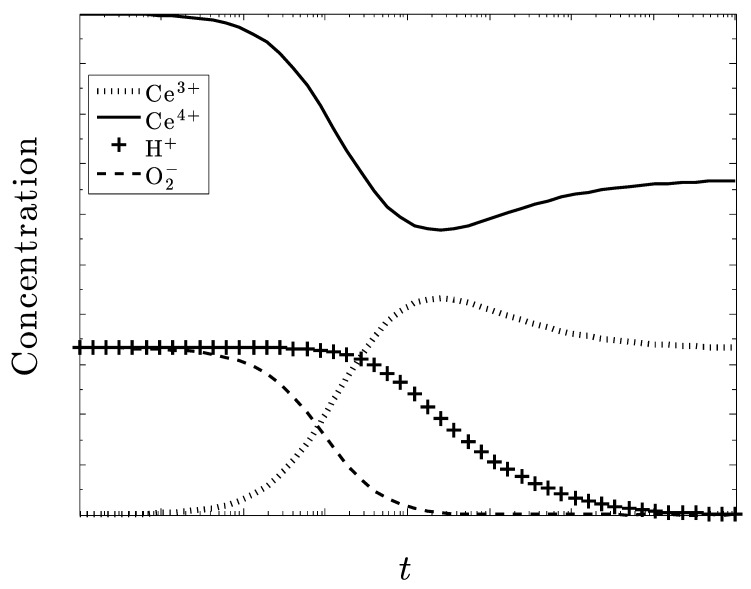
An increased dose of CeO_2_ that is three times the amount of superoxide. The concentration of superoxide ions is rapidly eliminated, while leaving a balance of Ce^3+^ that is not depleted.

**Figure 7 biomolecules-09-00447-f007:**
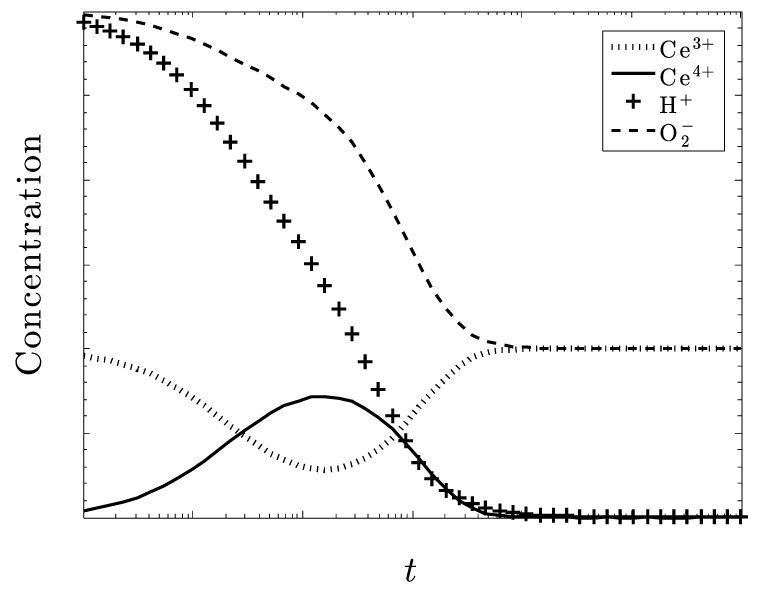
An initial concentration of superoxide three times the concentration of ceria, all in the +3 state. The concentration of superoxide ions is not well controlled.

**Figure 8 biomolecules-09-00447-f008:**
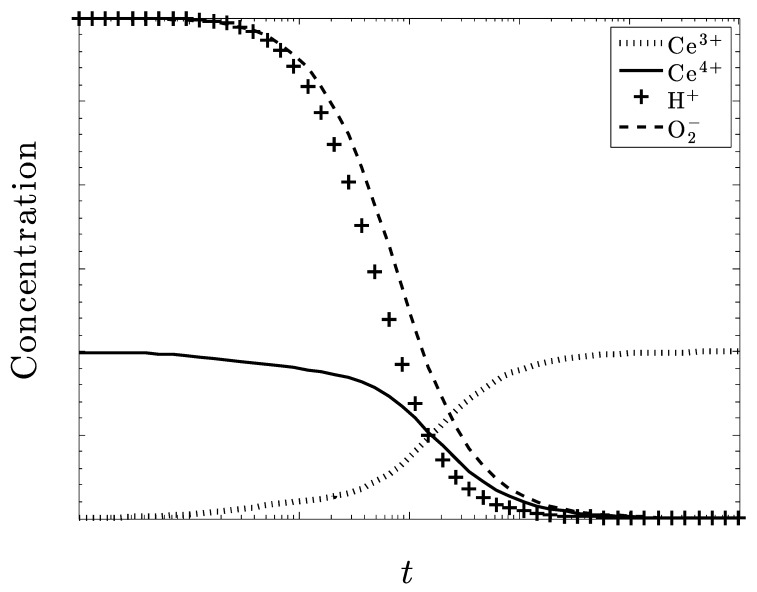
The concentration of superoxide ions is three times the concentration of ceria, now all in the +4 state. The concentration of superoxide is now well controlled.

**Figure 9 biomolecules-09-00447-f009:**
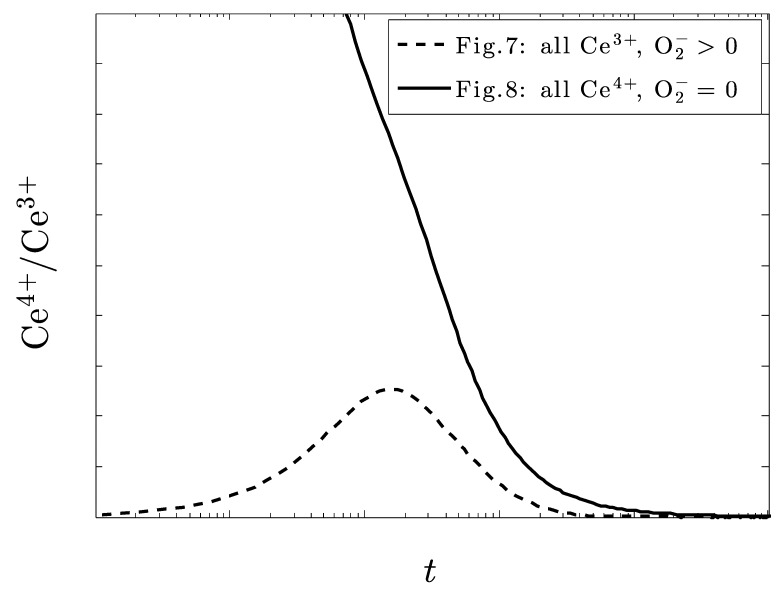
The calculated ratio of Ce^3+^/Ce^4+^corresponding to the situation in Figure 7 (all Ce^3+^ initially) and Figure 8 (all Ce^4+^) when the system is overrun with superoxide.

**Figure 10 biomolecules-09-00447-f010:**
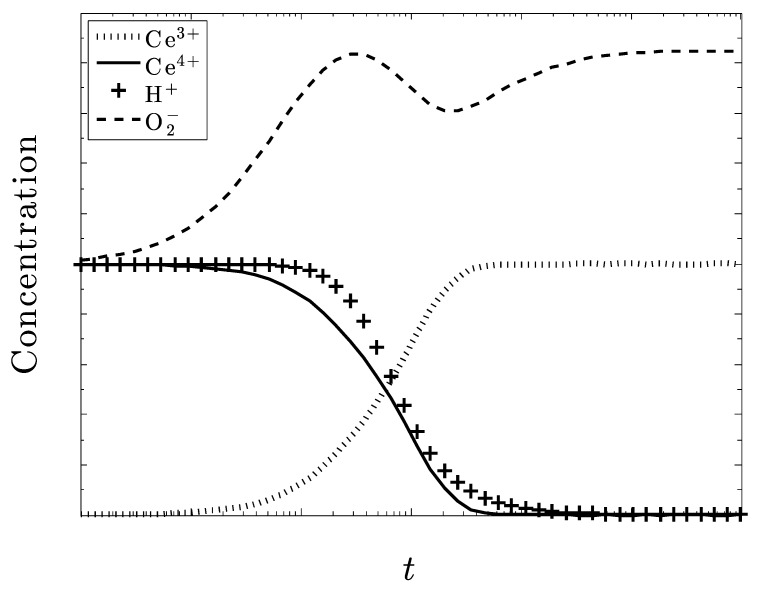
In this case, the amount of hydroxide is triple the amount of the other species. With all the rate constants set equal to each other, the amount of superoxide increases from its initial amount.

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
