# Peer review of "Modeling the Kinetic Behavior of Reactive Oxygen Species with Cerium Dioxide Nanoparticles"

_biomolecules, 2019, doi:10.3390/biom9090447_

Round 1

Reviewer 1 Report

I have seen the revised paper and in my opinion is more complete and deserves to be published in the present form

Author Response

Thanks for your comments

Reviewer 2 Report

All the requested modifications were applied.

There is only something strange concerning the organization of the text: there are two sections "Results and discussion" (they should be one!), one additional section "Discussion" and then "Conclusions". The last two could become a more extensive "Conclusions" paragraph, in my opinion.

Author Response

Suggested changes were made

Reviewer 3 Report

The authors replied to the reviewer's questions. The nomenclature of the symbols can be improved to avoid confusions. The authors exchange "O" and "0" many times when they described the chemical formulae related to ceria. Also. there are again typos or mixed up when the authors stated "ceria" in place of "cerium). See line 326. 

Author Response

In the body of the text these changes have been made for clarity but in the equations  the equation editor presents the letter O in a somewhat enlongated form that looks like a zero.

This is a font quirk of the editor and cannot be changed but someone reading the equations would automatically assume the symbol was that of oxygen and not a zero since one does not differentiate a zero ie d0/dt does not make any sense but dO/dt does.

This manuscript is a resubmission of an earlier submission. The following is a list of the peer review reports and author responses from that submission.

Round 1

Reviewer 1 Report

 I do recommend minor revision taking into account that in my opinion not all the data are well organized and therevision is a need to take into account the following:.

1 Kinetics and mathematical treatment need quantification and data not only in figures, but tables as well.

2 words are presented with many words each of them and some of them represent the same thing (see ROS and reactive oxygen species )

Reviewer 2 Report

It was interesting to me to read this paper dealing with the variables affecting the CeO2 reaction and the connection with the formation of ROS. I found the topic interesting for people working with nanometric CeO2 for medical applications. I suggest the publication with minor modifications. 

The points not clear to me were the following:

1) in Figure 1 I do not understand the meaning of the arrows. This aspect whould be integrated in the figure caption.

2) In the equation in line 140 D and H were not defined.

3) It was not convincing the division of the paragraphs in "Results, steady state solutions", "Results, time-dependent kinetics" and "Discussion". I suggest to reorder into "Results and discussion" section with two subsections "Steady state solutions" and "time-dependent kinetics". Moreover, the actual "Discussion" section seems more a summary of the results than a real discussion part, I suggest to have a careful look to a possible reorganization of the text. 

4) In the captions of Fig 2 to Fig 10 some sentences reporting some sort of conclusions appear (for instance in Fig.2 "The ceria is antagonistic and somewhat ineffective as a medical treatment." or in Fig.3 "The system is very active in controlling superoxide unlike the results in Figure 2."). I suggest to avoid this and to discuss the trends of the curves in the Results (and discussion) section. 

Reviewer 3 Report

The manuscript reports a mathematical model to describe the reaction kinetics between two selected reactive oxygen species and cerium oxide particles. Currently, most of the reported research in ceria focuses on “look and see” experiments. Thus, the materials in the manuscript provide a nice refreshing perspective of the subject. Particularly, the self-regulating and self-limiting concepts are nicely pointed out. Nevertheless, this manuscript lacks sufficient considerations of the complex reactions in this catalytic process. There are a lot of implicit assumptions in the model. As a theoretical study, this is fine if the majority of them are specifically discussed and laid out clearly. Below are detailed comments.

1. The manuscript assumes that most reactions involve only superoxide and hydrogen peroxide. It is clear that there are many other species such as hydroxyl radicals and other ROS species involve in the reaction. Such assumptions should be discussed specifically. Also, the authors assume complete availability of all cerium species as free ions in the model. This is not the case in real ceria samples. As a matter of fact, the ceria lattice structure allows the fast redox reactions on ceria’s surface.

2. Some of the elementary reactions (line 76) do not agree with each other. This led to errors in the model. For example, it does not make sense to involve both hydroxide ions and hydronium ions as major species in the elementary reactions. This led to the unreasonable assumption in the discussion (line 328) that the ionization of water is not important in the model. The reviewer encourages the authors to redo the modeling with this considered when they solve the differential equations. The authors also can easily resolve it by adding water to both sides of the reaction equation (1). If this is really a minor issue in the modeling, the authors should present evidence to support this point.

3. Line 112: It should be “cerium ions ratio” not “ceria ratio”.

4. Some of the labeled symbols were not mentioned specifically. For example, D represents the concentration of hydroxide ions. This should be rechecked thoroughly in the manuscript.